# Sequestration of host metabolism by an intracellular pathogen

Lena Gehre[1,2], Olivier Gorgette[3], Stéphanie Perrinet[1,2], Marie-Christine Prevost[3], Mathieu Ducatez[4], Amanda M Giebel[5], David E Nelson[6], Steven G Ball[4], Agathe Subtil[1,2]*

[1]Unité de Biologie cellulaire de l'infection microbienne, Institut Pasteur, Paris, France; [2]CNRS UMR3691, Paris, France; [3]Plate-forme de Microscopie Ultrastructurale, Imagopole, Institut Pasteur, Paris, France; [4]Unité de Glycobiologie Structurale et Fonctionnelle - CNRS UMR8576, Université de Lille, Lille, France; [5]Department of Biology, Indiana University Bloomington, Bloomington, United States; [6]Department of Microbiology and Immunology, Indiana University School of Medicine, Indianapolis, United States

**Abstract** For intracellular pathogens, residence in a vacuole provides a shelter against cytosolic host defense to the cost of limited access to nutrients. The human pathogen *Chlamydia trachomatis* grows in a glycogen-rich vacuole. How this large polymer accumulates there is unknown. We reveal that host glycogen stores shift to the vacuole through two pathways: bulk uptake from the cytoplasmic pool, and *de novo* synthesis. We provide evidence that bacterial glycogen metabolism enzymes are secreted into the vacuole lumen through type 3 secretion. Our data bring strong support to the following scenario: bacteria co-opt the host transporter SLC35D2 to import UDP-glucose into the vacuole, where it serves as substrate for *de novo* glycogen synthesis, through a remarkable adaptation of the bacterial glycogen synthase. Based on these findings we propose that parasitophorous vacuoles not only offer protection but also provide a microorganism-controlled metabolically active compartment essential for redirecting host resources to the pathogens.

*For correspondence: asubtil@pasteur.fr

## Introduction

Many microorganisms develop inside eukaryotic cells, either free in the cytosol or enclosed in a vacuole (*Creasey and Isberg, 2014*; *Fredlund and Enninga, 2014*). Each microorganism adapts its intracellular metabolism to the nutrient supply of the host (*Abu Kwaik, 2015*; *Eisenreich et al., 2010*). One recognized advantage of a vacuole is that it provides a shelter against cytosolic host defense (*Kumar and Valdivia, 2009*), to the cost of limited access to cytosolic nutrients. Acquisition of nutrients through this barrier is a crucial feature of host-microbe interaction.

*Chlamydiae* are Gram-negative obligate intracellular bacteria found as symbionts and pathogens in a wide range of eukaryotes, including protists, invertebrates and vertebrates (*Horn, 2008*). The developmental cycle of *Chlamydiae* involves two morphologically distinct forms. Infectious particles, called elementary bodies (EBs), are small and adapted to extracellular survival. After invasion of the host cell, they establish a parasitophorous vacuole called an inclusion, and convert within the first hours into larger organisms with higher metabolic activity, called reticulate bodies (RBs). RBs replicate several times within the inclusion, until they differentiate back into EBs, in a non synchronous manner (*AbdelRahman and Belland, 2005*). The human adapted strain *C. trachomatis* is the leading cause of infectious blindness (*Taylor et al., 2014*) as well as of sexually transmitted infections caused

**eLife digest** *Chlamydia trachomatis* is the most common sexually transmitted bacteria that causes disease. Infections often do not produce any obvious symptoms, but can lead to infertility or other severe problems if left untreated. This microbe is also the leading cause of blindness by an infectious agent.The bacteria grow in the human body by infecting host cells. Inside these cells, the bacteria are found inside compartments known as inclusions, which protect them from the host's defense responses and enable them to create a comfortable environment for themselves. However, this comes at a cost because the bacteria lose immediate access to the nutrients in the rest of the host cell. Thus, *C.trachomatis* has developed ways to import these nutrients into inclusions, and, more generally, to take the control of its interactions with the host cell.

The inclusions built up by *C. trachomatis* contain a high amount of glycogen, a carbohydrate that generally acts as an energy storage molecule. Although this observation was made many decades ago, the molecular mechanism by which such a large molecule accumulates in the inclusion has not been clarified. Gehre et al. have now used a variety of cell biology techniques to address this question.

The experiments show that there are two different pathways through which glycogen accumulates within the inclusion. Some glycogen is transported in bulk from the interior of the host cell into the inclusion. However, the bacteria also make new glycogen in the inclusion from a building block molecule called UDP-glucose. To do this, the bacteria recruit a host transport molecule to the membrane that surrounds the inclusion. This transport molecule brings UDP-glucose into the inclusion, where an enzyme called glycogen synthase – which is released by the bacteria – uses the UDP-glucose to make glycogen. The *C. trachomatis* glycogen synthase is unusual because most other bacteria can only make glycogen from another type of glucose.

By using both pathways, *C. trachomatis* is able to trap most of the glycogen stores of the infected cell within the inclusion so that they are inaccessible to the host but ready for the bacteria to use. Previous work has shown that *C. trachomatis* is much better at accumulating glycogen than other *Chlamydia* bacteria are. Therefore, a future challenge will be to find out exactly how this helps *C. trachomatis* survive inside human cells.

by bacteria. Infections of the urogenital mucosae often stay asymptomatic causing irreparable damage leading to ectopic pregnancies or tubal factor infertility (*Brunham and Rey-Ladino, 2005*).

C. *trachomatis* displays a genome reduced to around one million base pairs and therefore highly relies on the host with regard to several essential metabolic pathways, such as nucleotide or amino acid biosynthesis (*Stephens et al., 1998*). Lipid droplets and peroxisomes have been observed in the inclusion lumen, indicating that this compartment is able to engulf large particles to meet bacterial needs (*Boncompain et al., 2014*; *Kumar et al., 2006*). C. *trachomatis,* and the closely related mouse and hamster pathogen *C. muridarum,* are unique amongst *Chlamydiae* for their ability to accumulate glucose (Glc) under the form of glycogen in the inclusion lumen (*Gordon and Quan, 1965*). Glc deprivation leads to a complete loss of infectivity (*Harper et al., 2000*; *Iliffe-Lee and McClarty, 2000*). The *C. trachomatis* genome encodes for all the enzymes necessary for a functional glycogenesis and glycogenolysis (*Stephens et al., 1998*) (*Figure 1A*). Although they do not accumulate glycogen in the inclusion, other *Chlamydia* species also have a complete set of those enzymes, a surprising observation given that this pathway is absent in most intracellular bacteria (*Henrissat et al., 2002*). Intriguingly, pathogenic *Chlamydiae* lack a hexokinase, the enzyme phosphorylating Glc, and therefore rely on the import of phosphorylated sugars for glycolysis or glycogenesis. During bacterial glycogenesis, Glc-1-phosphate (Glc1P) serves as a substrate for the enzyme ADP-Glc pyrophosphorylase (GlgC) giving rise to ADP-Glc, which in turn is the building block for a linear chain of $\alpha$1,4-linked Glc molecules. Branching of chains through $\alpha$1,6-glucosidic bond is introduced by the branching enzyme GlgB, giving rise to glycogen. The glycogen phosphorylase GlgP, the debranching enzyme GlgX and the amylomaltase MalQ are involved in the degradation of the glycogen particle to Glc1P (*Colleoni et al., 1999*; *Seibold and Eikmanns, 2007*; *Wilson et al., 2010*). Finally, the very large majority of circulating *C. trachomatis* strains contain a 7.5

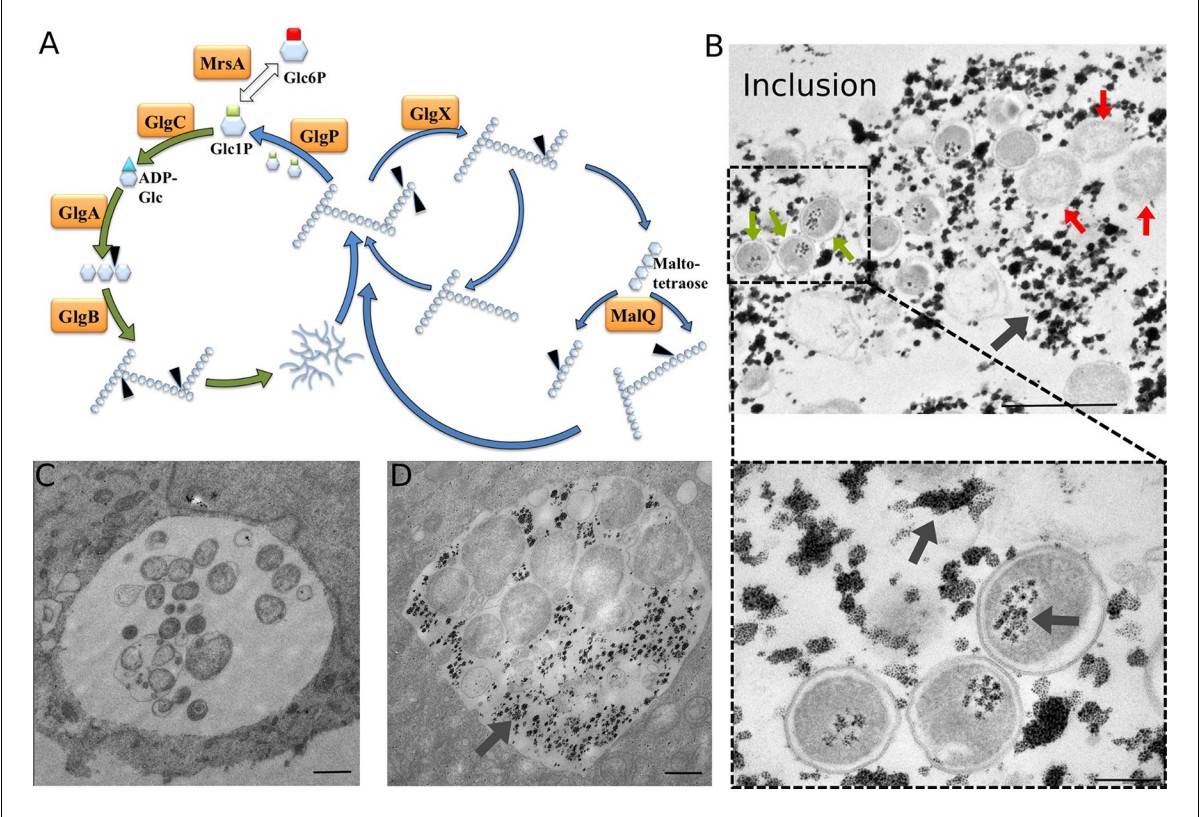

**Figure 1.** Glycogen accumulation in *C. trachomatis* inclusions. (**A**) Glycogen metabolism in bacteria. In green: glycogen synthesis. In blue: glycogen degradation. Glc1P is the substrate of GlgC for ADP-Glc synthesis. GlgA (glycogen synthase) produces linear glycogen chains via α-1,4 glycosidic bonds, and the branching enzyme (GlgB) introduces branches through α-1,6 linkages. Glycogen depolymerization in Glc1P is the result of the activity of GlgP (glycogen phosphorylase), GlgX (debranching enzyme) and MalQ (4-α-glucanotransferase). The phosphoglucomutase MrsA converts Glc1P to Glc6P. The arrows point to the site within the polysaccharide that is subjected to enzymatic activity. Genes for all these enzymes are present in *C. trachomatis.* (**B**) HeLa cells were infected for 30 hr with *C. trachomatis.* Glycogen was visualized by TEM after PATAg stain. Grey arrows point at glycogen. Green arrows point at examples of EBs and red arrows at RBs. The picture on the bottom shows an enlargement of the boxed region containing three EBs. Scale bars: 1 µm (top), 200 nm (bottom). (**C,D**) Cells were Glc-deprived 48 hr prior to infection. 10 mg/ml Glc were added 24 hpi and cells were (**C**) fixed immediately or (**D**) 4 hr after Glc administration. Note that no glycogen is detectable in the bacteria while it is highly abundant in the inclusion lumen. Scale bar: 1 µm.

The following figure supplements are available for figure 1:

**Figure supplement 1.** Relocation of glycogen stores to the inclusion during infection.

**Figure supplement 2.** Luminal and cytoplasmic glycogen differs in size.

**Figure supplement 3.** Kinetics of glycogen accumulation.

kb plasmid. Loss of this plasmid is associated with decreased GlgA expression and impaired glycogen accumulation (*Carlson et al., 2008*).

It is widely believed that the abundance of polysaccharides in the inclusion reflects the accumulation of glycogen in the bacteria themselves. While luminal, extrabacterial, glycogen has been observed (*Chiappino et al., 1995*), its origin has not been investigated. Here we show that luminal glycogen is in fact derived from bulk import of host glycogen and also *de novo* intraluminal synthesis, through the action of secreted bacterial enzymes. We reconstruct Glc metabolism in infected cells and demonstrate the ability for a microbe to convert its vacuole lumen into a compartment for regulated metabolic activity.

## Results

### Glycogen accumulation in the inclusion lumen is not the result of bacterial lysis

To determine whether glycogen accumulated in the host cytosol, in the inclusion, or both, we labeled polysaccharides using periodic-acid-Schiff (PAS) staining at different times of infection. Large glycogen particles were detected in most non-infected cells (*Figure 1—figure supplement 1*). Twenty-four hours post infection (hpi), glycogen was still detected in the cytoplasm of some of the infected cells, and the inclusions only showed weak PAS staining. However, 48 hpi no glycogen particle was detected in the cytoplasm of infected cells, while inclusions heavily stained with PAS, indicating that the global increase in glycogen content is accompanied by a shift in its original cytosolic localization, in favor of the bacterial inclusion. We used transmission electron microscopy (TEM) to determine more precisely its subcellular localization and detected the polysaccharide in two locations: in the inclusion lumen, and within EBs (*Figure 1B*). Intraluminal glycogen deposits are physically larger than in the cytoplasm (*Figure 1—figure supplement 2*). We did not observe glycogen in RBs, in contrast to an earlier report (*Chiappino et al., 1995*). In that publication, the presence of glycogen in the inclusion lumen was interpreted as the result of glycogen release from lysed bacteria. Considering the abundance of glycogen in the inclusion lumen relative to its amount in bacteria we considered this hypothesis unlikely. We tested it by depriving the cells of Glc for 48 hr before infecting them. Under these conditions, the inclusions contained no glycogen 24 hpi (*Figure 1C*). Restoring Glc availability for 4 hr was sufficient to trigger the accumulation of glycogen in the inclusion lumen, but not in the bacteria (*Figure 1D*). This experiment demonstrates that glycogen in the inclusion lumen does not result from the release of bacterial stores. The kinetics of glycogen appearance in the inclusion was carefully examined next by TEM. Luminal glycogen first appeared between 16 and 20 hpi (*Figure 1—figure supplement 3*), and was abundant 24 hpi. Thus, luminal glycogen deposit took place at a time when RBs largely predominate over EBs. This observation suggests that, while glycogen accumulation in the inclusion is most obvious at later stages of infection, when EBs accumulate, the process is initiated by RBs.

### Part of luminal glycogen is translocated in bulk from the host cytoplasm

Our conclusion raised an obvious question: how could a large polymer appear in the inclusion lumen? Two mechanisms are conceivable: bulk translocation of host glycogen, or transport of monomeric substrates (such as nucleotide-sugars or hexose phosphates) across the inclusion membrane for *de novo* luminal polymerization. Gys1, the host glycogen synthase, produces linear chains of Glc and is tightly bound to glycogen (*Luck, 1961*; *Stapleton et al., 2010*). Gys1 staining with specific antibodies gave a patchy pattern in the cytoplasm, as expected since it mirrors the distribution of glycogen. In addition, we observed that the enzyme accumulated within the inclusion lumen. Staining was specific since it disappeared when Gys1 expression was knocked down using siRNA prior to infection (*Figure 2A*). Observation of cytoplasmic markers in the inclusion can result from post-fixation artifact (*Kokes and Valdivia, 2015*). When we deprived the cells of Glc for two days before infection, and thus decreased cytoplasmic glycogen, Gys1 staining in the inclusion lumen was considerably reduced, while remaining abundant in the cytoplasm. This observation shows that Gys1 appearance in the inclusion lumen requires the presence of host glycogen, and is likely not the result of a post-fixation artifact. (*Figure 2—figure supplement1*). Altogether these data favored the scenario of bulk import of host glycogen and glycogen-bound Gys1 into the inclusion. TEM observations also came in support of this scenario, since we could observe one or more glycogen-filled vesicular structures in the inclusion lumen of about 20% of the glycogen-positive inclusions we examined (*Figure 2B*). Since in some cases several membranes were observed around luminal glycogen, we hypothesized that cytoplasmic glycogen might be entrapped by an autophagy-dependent process in the host cytoplasm, before delivery to the inclusion lumen. To test this hypothesis we used an *Atg5-/-* mutant mouse embryonic fibroblast (MEF) cell line, which is deficient in autophagy (*Kuma et al., 2004*). Inclusions of *Atg5-/-* MEFs still harboured glycogen, Gys1 and glycogen-filled vesicles, demonstrating that the pathway of bulk host glycogen uptake is independent of autophagy (*Figure 2—figure supplement 2*).

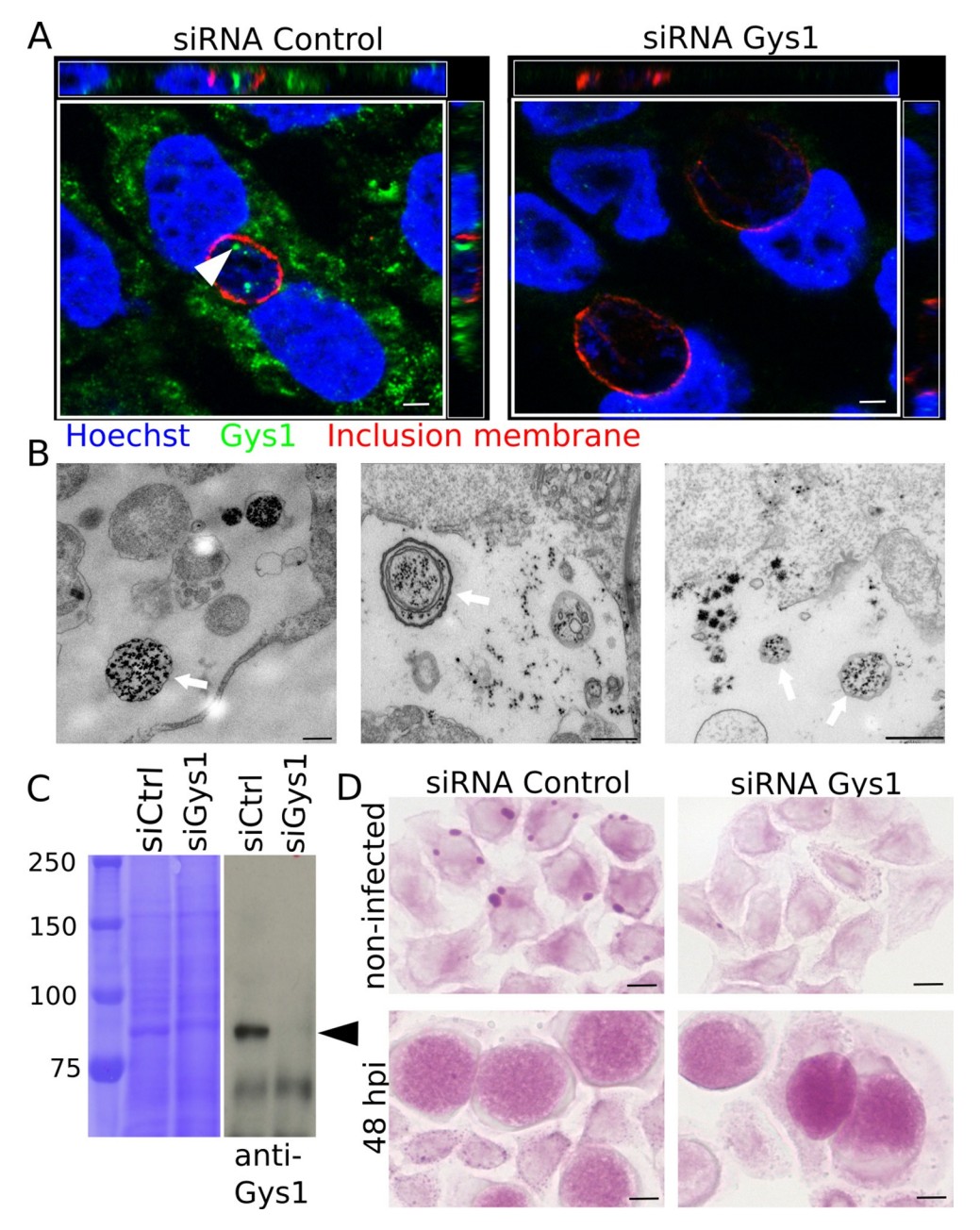

**Figure 2.** Bulk import of cytoplasmic glycogen contributes to the accumulation of glycogen in the inclusion. (A) Gys1 is imported into the inclusion lumen. Cells were treated with siRNA control or against Gys1 prior to infection for 30 hr. DNA was stained in blue, Gys1 in green and the inclusion membrane in red (anti-CT813). The white arrowhead points to intraluminal Gys1, see also xz (top) and yz (right) projections. Scale bar: 10 µm. (B) Examples of TEM images of glycogen-filled vesicles (arrows) in inclusions 30 hpi. Glycogen is visualized through PATAg staining. Scale bar: 500 nm. (C, D) Cells were treated with either siRNA control or siRNA against Gys1 48 hr prior to infection. (C) Coomassie staining of whole cell lysates as loading control (left) and immunoblot with an anti-Gys1 antibody (right). The arrowhead points to Gys1 (MW = 85 kDa). (D) PAS on non-infected cells fixed 48 hr after siRNA treatment (top) and on siRNA treated cells infected for 48 hr (bottom). Scale bar: 10 µm.

The following figure supplements are available for figure 2:

**Figure supplement 1.** Luminal location of Gys1 strongly decreases in glucose-deprived cells.

**Figure supplement 2.** Host glycogen imported in bulk from the host is not of autophagic origin.

# Chlamydial glycogen enzymes are secreted into the inclusion lumen for *de novo* glycogen synthesis

If bulk glycogen import is the only mechanism at work, depleting host glycogen stores should dramatically reduce intraluminal glycogen content. Cells were transfected with siRNA against Gys1 two days before infection. Silencing was efficient (*Figure 2C*) and there was little glycogen left in the host cytoplasm at the time of infection (*Figure 2D*, top). However, knocking down Gys1 expression did not abolish glycogen accumulation in the inclusions (*Figure 2D*, bottom). We measured the pixel intensity of the PAS staining in inclusions 48 hpi, and used inclusion staining of cells infected in the absence of Glc to determine the background level, which we subtracted. Inclusions of cells treated with siRNA against Gys1 contained 82% (s.e.m. 4.9) of the glycogen measured in cells treated with control siRNA. This experiment strongly suggested that bulk import of host glycogen was at least partially responsible for intraluminal glycogen accumulation. However, it did not appear to be the main contributor and indicated that *de novo* glycogen synthesis in the inclusion lumen also took place.

Synthesis of glycogen, and possibly its degradation into Glc monomers amenable to bacterial uptake, implies that glycogen synthesis and degradation enzymes are present in the inclusion lumen. While Gys1 import in the inclusion lumen might contribute, it cannot account for the glycogen accumulation observed in Gys1 depleted cells. We measured the expression profile of several proteins involved in glucose metabolism, together with control genes. For most proteins, RNA levels increased between 8 and 24 hpi, mirroring the known increase in bacterial metabolic activity in this time window (*Figure 3—figure supplement 1*). *glgA* stood out, as its transcription was delayed until 16 hpi. *glgA* expression thus correlates very well with the timing of glycogen appearance in the inclusion lumen, between 16 and 20 hpi, implicating that the enzyme might be involved in luminal glycogen synthesis. Indeed, secretion of GlgA into the inclusion lumen, as well as into the host cytoplasm, has recently been demonstrated (*Lu et al., 2013*). In addition, a GlgB mutant strain shows massive precipitation of glycogen in the inclusion (*Nguyen and Valdivia, 2012*), due to accumulation of unbranched glycogen, implicating that GlgB normally functions in the inclusion lumen. These data strongly support the hypothesis that bacterial enzymes are at the origin of glycogen synthesis in the inclusion lumen. To determine if bacterial enzymes also take over glycogen depolymerization we obtained a specific antibody against the chlamydial debranching enzyme GlgX. We controlled the specificity of the staining by western blot and by immunofluorescence (*Figure 3—figure supplement 2*). In cells infected for 24 hpi or 48 hpi, GlgX was found in the inclusion lumen, with mostly no overlap with bacteria, demonstrating secretion within the inclusion lumen (*Figure 3A*). Interestingly, GlgX was also detected on the inclusion membrane 24 hpi, but not 48 hpi (*Figure 3A and Figure 3—figure supplement 3*). Altogether, these data show that bacterial enzymes for glycogen synthesis and depolymerisation are present in the inclusion lumen, where they likely control glycogen polymerization/depolymerization.

How are these bacterial enzymes transported to the inclusion lumen? None of them contain a signal peptide for transit to the periplasm and further transport by the type 2 or 5 secretion pathway (SignalP 4.1). Since type 3 secretion is the prominent pathway for protein secretion in *Chlamydia* (*Betts et al., 2009*) we looked for the presence of a type 3 secretion (T3S) signal (*Galan et al., 2014*) in the amino terminal domain of all glycogen metabolism enzymes. The N-termini of the proteins of interest were fused to a reporter (calmodulin-dependent adenylate cyclase of *Bordetella pertussis* Cya), and the secretion of the resulting chimera was evaluated in *Shigella flexneri*, in an assay previously validated (*Subtil et al., 2005*; *Subtil et al., 2001*). Five out of the six chimera tested, GlgA/Cya, GlgB/Cya, GlgX/Cya, GlgP/Cya and MalQ/Cya, were secreted in the supernatant of cultures when transformed in a *S. flexneri* mutant with constitutive T3S (*ipaB* strain), and not when transformed in a mutant deficient for T3S (*mxiD* strain). The endogenous T3S substrate of *S. flexneri*, IpaD, was also secreted only in the *ipaB* background (positive control), while cAMP receptor protein (CRP), a non-secreted protein, was found exclusively in the bacterial pellet, excluding the possibility of non-specific leakage into the supernatant. The sixth chimera tested, GlgC/Cya, was not detected in the culture supernatant, suggesting that GlgC, which converts Glc1P into ADP-Glc, is not a substrate of T3S (*Figure 3B*).

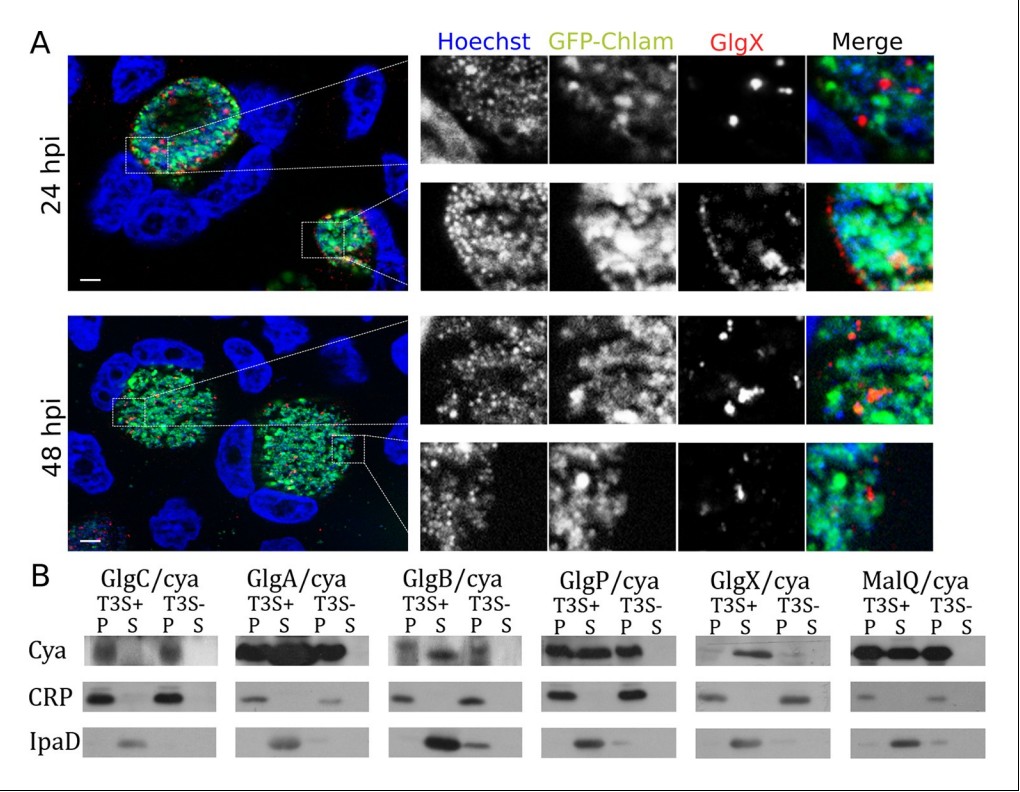

**Figure 3.** Bacterial glycogen metabolism enzymes are secreted in the inclusion lumen. (**A**) HeLa cells were infected for 24 hr or 48 hr with GFP expressing L2 (GFP-Chlam, green) and stained with an anti-GlgX antibody (red) and Hoechst (blue). Insets to the right show enlargements of the boxed areas. Scale bar: 5 μm. (**B**) Heterologous test of secretion: the N-terminal 20 amino acids of the indicated proteins were fused to the reporter Cya, and constructs were transformed into the *S. flexneri* strains *ipaB* (T3S+) and *mxiD* (T3S-). Liquid cultures were fractionated into pellet (P) and supernatant (S). All chimeras except GlgC/cya were detected in the supernatant in T3S competent bacteria and not in T3S defective bacteria, indicative of a functional T3S signal. The endogenous T3S substrate of *S. flexneri*, IpaD, serves as a positive control, while the non-secreted cAMP receptor protein (CRP) controls for non-specific leakage into the supernatant.

The following figure supplements are available for figure 3:

**Figure supplement 1.** Expression profiles of genes related to glycogen metabolism.

**Figure supplement 2.** Specificity of the anti-GlgX antibody.

**Figure supplement 3.** GlgX accumulates at the inclusion membrane.

Collectively, these data strongly support the second scenario we proposed: chlamydial enzymes for glycogen metabolism are secreted by a T3S dependent mechanism into the inclusion lumen, and account for the glycogen accumulation observed even in the absence of host glycogen.

### *Chlamydia* import Glc6P, not Glc1P nor Glc

Secretion of the chlamydial glycogen degradation enzymes, GlgX, GlgP and MalQ, is consistent with a decrease in luminal glycogen content observed at late infection stages (*Gordon and Quan, 1965*), likely to feed EBs monomeric sugars (*Omsland et al., 2012*). We investigated what form of Glc the bacteria are able to take up by incubating purified EBs with radioactively labeled [C14]-Glc, [C14]-Glc1P or [C14]-Glc6P, in absence or presence of a 50-fold excess of non-radioactive Glc, Glc1P or Glc6P. Only [C14]-Glc6P was taken up by bacteria, and only Glc6P could compete it out (*Figure 4A*), demonstrating that the uptake was saturable, as expected for a transporter. There is only one

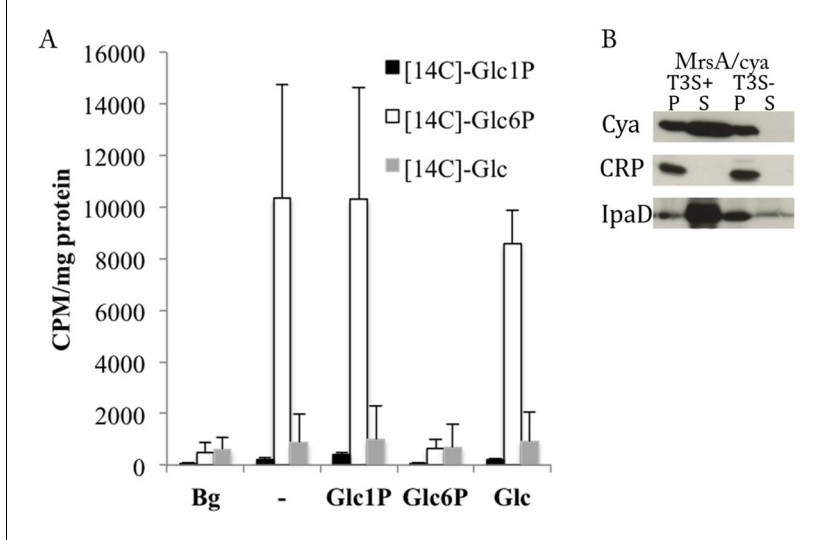

**Figure 4.** *C. trachomatis* takes up Glc6P and secretes phosphoglucomutase (MrsA). (**A**) Purified EBs were incubated for 2 hr with [14C]-Glc, [14C]-Glc6P or [14C]-Glc1P in absence or presence of a 50-fold excess of non-radioactive Glc, Glc6P or Glc1P. Bacteria were subsequently washed and pelleted and radioactivity measured. Background (Bg) was measured after 1 min of incubation instead of 2 hr. CPM: counts per minute. Results show mean and S.D. (n=3). (**B**) Heterologous test of secretion was performed on MrsA as in *Figure 3B*.

annotated hexose phosphate transporter in *C. trachomatis* (UhpC), which is likely responsible for Glc6P uptake, as the homologous protein in *C. pneumoniae* showed such activity (*Schwoppe et al., 2002*). Therefore there is a discrepancy between the glycogen degradation product, Glc1P, and the substrate of import into the bacteria, Glc6P. To solve this conundrum we hypothesized that Glc1P might be converted into Glc6P in the inclusion lumen. *C. trachomatis* encodes for a phosphogluco-mutase (MrsA), an enzyme allowing the interconversion between Glc1P and Glc6P. Using the heter-ologous test of secretion we found that *C. trachomatis* MrsA contains a functional T3S signal (*Figure 4B*). This suggests that Glc1P might be converted into Glc6P by MrsA inside the inclusion lumen, followed by uptake through UhpC.

## UDP-Glc is a substrate for chlamydial GlgA and is the host sugar transported into the inclusion lumen

Our data strongly support the hypothesis that chlamydial glycogen synthase (GlgA) initiates *de novo* glycogen synthesis in the inclusion lumen. This raises the question of its substrate specificity. A sharp distinction between prokaryote and eukaryote glycogen synthases is that, almost without exception, bacterial glycogen synthases function on ADP-Glc while eukaryotic glycogen synthases use UDP-Glc as substrate. We reasoned that GlgA secretion into the host cytoplasm, which contains no ADP-Glc, might point to an unusual substrate specificity for this enzyme. Transfection of cells with flag-tagged chlamydial GlgA led to massive glycogen accumulation, proving that indeed *C. trachomatis* GlgA is functional on UDP-Glc (*Figure 5A*). Interestingly, when transfected cells were subsequently infected, an increase in glycogen accumulation in the inclusion lumen was observed, and Flag-GlgA was detected in the bacterial compartment (*Figure 5—figure supplement 1*). High intraluminal glycogen content upon ectopic GlgA expression was also observed when cells were infected with a strain devoid of the natural plasmid of *C. trachomatis* (*Figure 5—figure supplement 1*), and Flag-GlgA was observed in the inclusion lumen also in that case (*not shown*). Plasmid-loss is associated with decreased GlgA expression and impaired glycogen accumulation (*Carlson et al., 2008*). In non-transfected cells the polysaccharide was still detected in EBs of the plasmid-less strain and, to a highly reduced level, in the inclusion lumen (*Figure 5—figure supplement 2*). Glycogen recovery upon GlgA transfection indicates that the low level of expression of GlgA in the plasmid-less strain accounts for the defect in glycogen storage. It is quite remarkable that a protein expressed by the host can compensate for a bacterial deficiency.

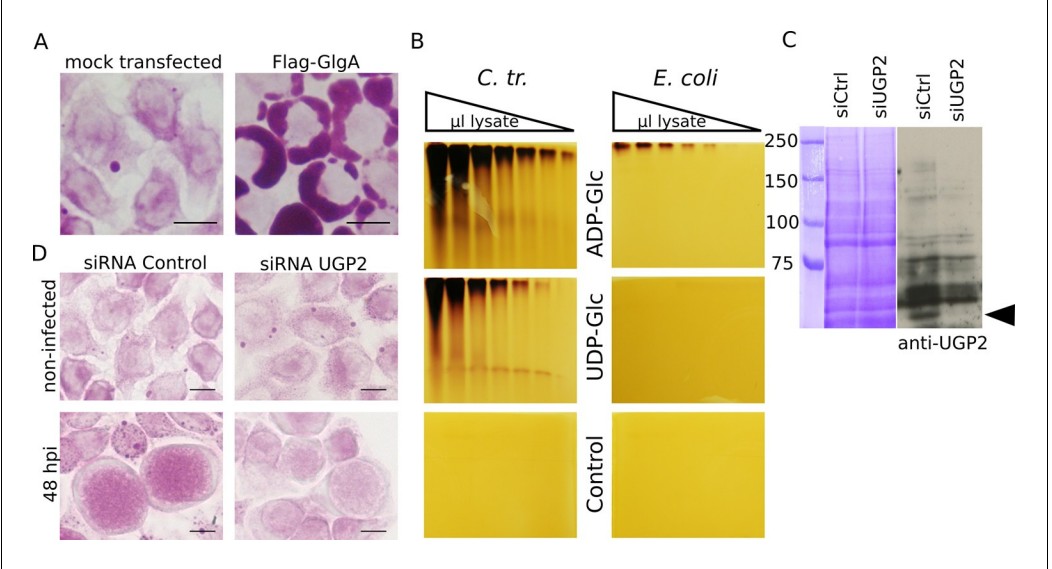

**Figure 5.** Host UDP-Glc is the substrate for intraluminal glycogen synthesis. (**A**) PAS staining was performed 24 hr after transfection with chlamydial Flag-GlgA. (**B**) Zymogram analysis. Serial dilutions of lysates of *E. coli* lacking endogenous *glgA* and transformed with chlamydial *glgA* (*C. tr.*) or *E. coli* *glgA* (*E. coli*) were separated by native polyacrylamide electrophoresis and incubated in either 1.2 mM ADP-Glc, UDP-Glc or buffer only (control). Glycogen production was visualized by iodine staining. (**C,D**) Cells were treated with either siRNA control or siRNA against UGP2 48 hr prior to infection. (**C**) Coomassie staining of whole cell lysates as loading control (left) and immunoblot with an anti-UGP2 antibody (right). The arrowhead points to UGP2 (MW = 56 kDa). (**D**) PAS on siRNA treated cells at time of infection (top) and at 48 hpi (bottom). Knocking down UGP2 expression results in the decrease of luminal glycogen accumulation.

The following figure supplements are available for figure 5:

**Figure supplement 1.** Flag-GlgA is imported into the inclusion lumen and enhances luminal glycogen accumulation.

**Figure supplement 2.** The plasmid-less strain accumulates glycogen in EBs and - to a lesser extent than the wild-type strain - in the inclusion lumen.

A zymogram analysis was performed to further compare chlamydial GlgA activity towards either UDP-Glc or ADP-Glc. Briefly, serial dilutions of lysates of *E. coli* lacking their endogenous *glgA* and transformed with either *E. coli* *glgA* or *C. trachomatis* *glgA* were separated on non-denaturing poly-acrylamide gels that contained rabbit glycogen. The gels were subsequently incubated in buffer containing either UDP-Glc or ADP-Glc, and glycogen production was visualized by iodine staining. While *E. coli* GlgA showed activity exclusively upon incubation with ADP-Glc, as expected, the chlamydial GlgA showed activity with both substrates (*Figure 5B*).

We next asked which Glc derivative might be translocated across the inclusion membrane for *de novo* glycogen synthesis. UDP-Glc stood as the best candidate: it is a substrate for GlgA, and its import would relieve the bacteria from the cost of nucleotide-sugar synthesis. In addition, GlgC (converting Glc1P into ADP-Glc) was the only glycogen metabolism enzyme that failed the *Shigella* T3S test, suggesting that this enzyme may not be present within the inclusion lumen. To test the requirement for UDP-Glc for luminal glycogen synthesis we silenced the expression of the host UDP-Glc pyrophosphorylase UGP2 (UGP2 catalyzes the conversion of Glc1P into UDP-Glc) by siRNA for two days before infection. We measured the pixel intensity of the PAS staining in inclusions as described above. Glycogen accumulation within the inclusion lumen decreased to 57% (s.e.m. = 5.8) in cells treated with a siRNA against UGP2 compared to cells treated with control siRNA (*Figure 5C,D*). These data strongly argue for UDP-Glc being the substrate for sugar import into the inclusion. Note that while UGP2 silencing also impacted host glycogen stores, as expected, it did so to a lesser extent than Gys1 silencing (*Figure 2D*). Thus the greater impact of UGP2 silencing on luminal glycogen stores cannot be explained by its indirect effect on bulk glycogen import. Instead it very likely reflects the requirement for UDP-Glc for *de novo* luminal glycogen synthesis.

## C. *trachomatis* co-opts host transporter SLC35D2 to import UDP-Glc into the inclusion lumen

We have demonstrated that *de novo* glycogen synthesis takes place in the inclusion lumen, initiated by chlamydial glycogen synthase GlgA, using UDP-Glc as substrate. This implies that UDP-Glc is translocated from the host cytoplasm into the inclusion lumen. Among the large family of annotated human sugar transporters, only SLC35D2 was experimentally shown to be able to transport UDP-Glc (*Suda et al., 2004*). In cells infected with *C. trachomatis*, a construct of the transporter bearing a C-terminal HA-tag was enriched around the inclusion (*Figure 6A*). By TEM, we detected the presence of the transporter on the inclusion membrane (*Figure 6B*). Staining was specific since we observed 0.96 gold particles per µm of inclusion membrane examined, against 0.12 gold particles per µm in non-transfected cells. The ability for SLC35D2 to reach the inclusion membrane indicates that it might play a role in UDP-Glc import. To test this hypothesis we silenced SLC35D2 expression using siRNA prior to infection. In these conditions, glycogen accumulation in the inclusion decreased to 43% (s.e.m. = 1.7) (*Figure 6C,D*), thus to a level similar to that when UGP2 had been silenced (*Figure 5C,D*). These data strongly suggest that *C. trachomatis* co-opts the host nucleotide-sugar transporter to favor UDP-Glc import into the inclusion lumen. Finally we wondered if by

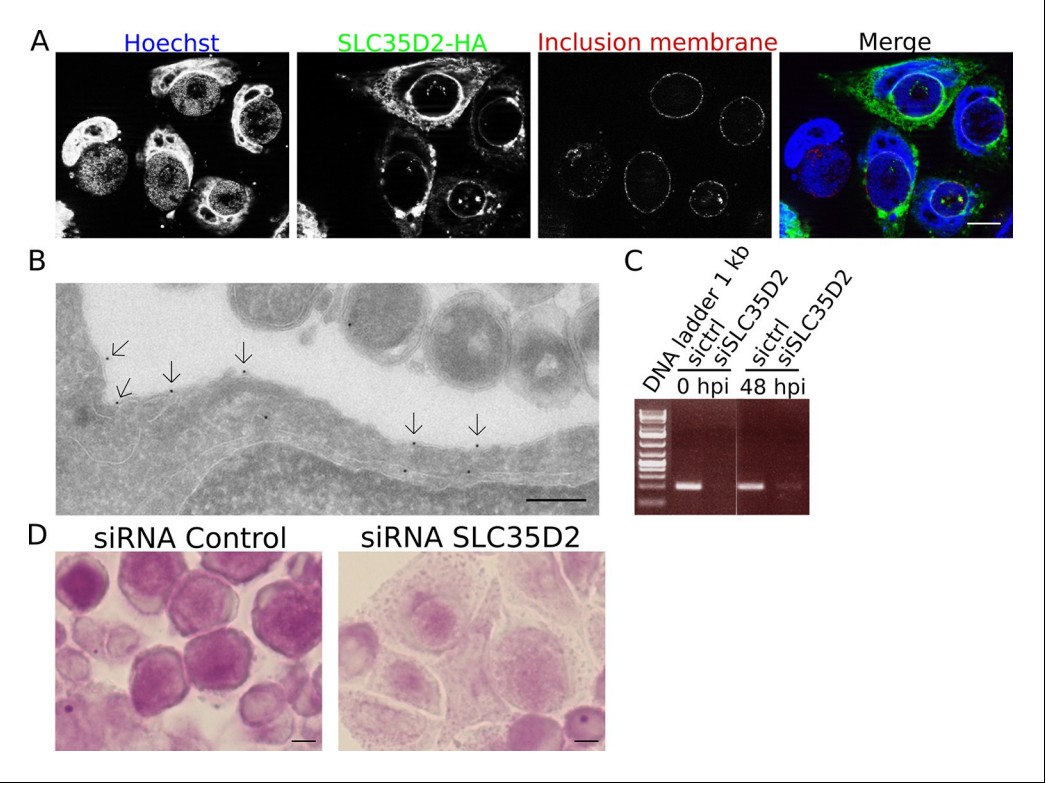

**Figure 6.** SLC35D2 imports UDP-Glc into the inclusion lumen. (**A, B**) Prior to infection cells were transfected with SLC35D2-HA, and fixed 24 hpi. (**A**) Labelling of the HA tag (green), inclusion membrane marker Cap1 (red) and DNA (blue) show recruitment of SLC35D2 to the inclusion periphery. Scale bar: 10 µm. (**B**) Duplicate samples were processed for TEM. Staining with anti-HA antibodies, followed with gold-coupled secondary antibodies, showed that SLC35D2-HA is detected on the inclusion membrane (arrows). Scale bar: 250 nm. (**C**) Cells were treated with siRNA control (ctrl) or siRNA SLC35D2 48 hr and 4 hr prior to infection. Samples were taken before infection (0 hpi) and 48 hpi, and RT-PCR was performed with primers specific to SLC35D2. (**D**) PAS staining of siRNA ctrl or siRNA SLC35D2 treated cells 48 hpi. Scale bar: 10 µm.

The following figure supplement is available for figure 6:

**Figure supplement 1.** Depletion of Gys1 does not further decrease luminal glycogen content in cells depleted for SLC35D2.

simultaneously impairing bulk glycogen import and UDP-Glc translocation we could further decrease luminal glycogen content. We depleted Gys1 and SLC35D2 for 48 hr before infecting the cells, and applied PAS staining 48 hr after infection. Additional Gys1 depletion did not decrease luminal glycogen content further than what SLC35D2 depletion did, confirming that in *C. trachomatis* infected cells bulk import of cytoplasmic glycogen plays only a minor part in intraluminal glycogen accumulation (*Figure 6—figure supplement 1A*). We measured the consequences on progeny 30 hpi to minimize the effect of SLC35D2 depletion on cell viability that we observed in some experiments. Decrease in progeny was never more than two-fold, indicating that even when both Gys1 and SLC35D2 were depleted simultaneously, there was no severe impact on bacterial development (*Figure 6—figure supplement 1B*).

## Mutation in GlgA results in defect in glycogen storage capacity and loss of infectivity

With the long-term goal of understanding the requirements for glycogen metabolism enzymes in a mouse model of infection we screened a mutant library of *Chlamydia muridarum* for strains with defects in glycogen accumulation. Most of the mutants obtained have a mutation in one of the enzymes for glycogen metabolism (Giebel et al, in preparation). We selected two clones, P2B10 and

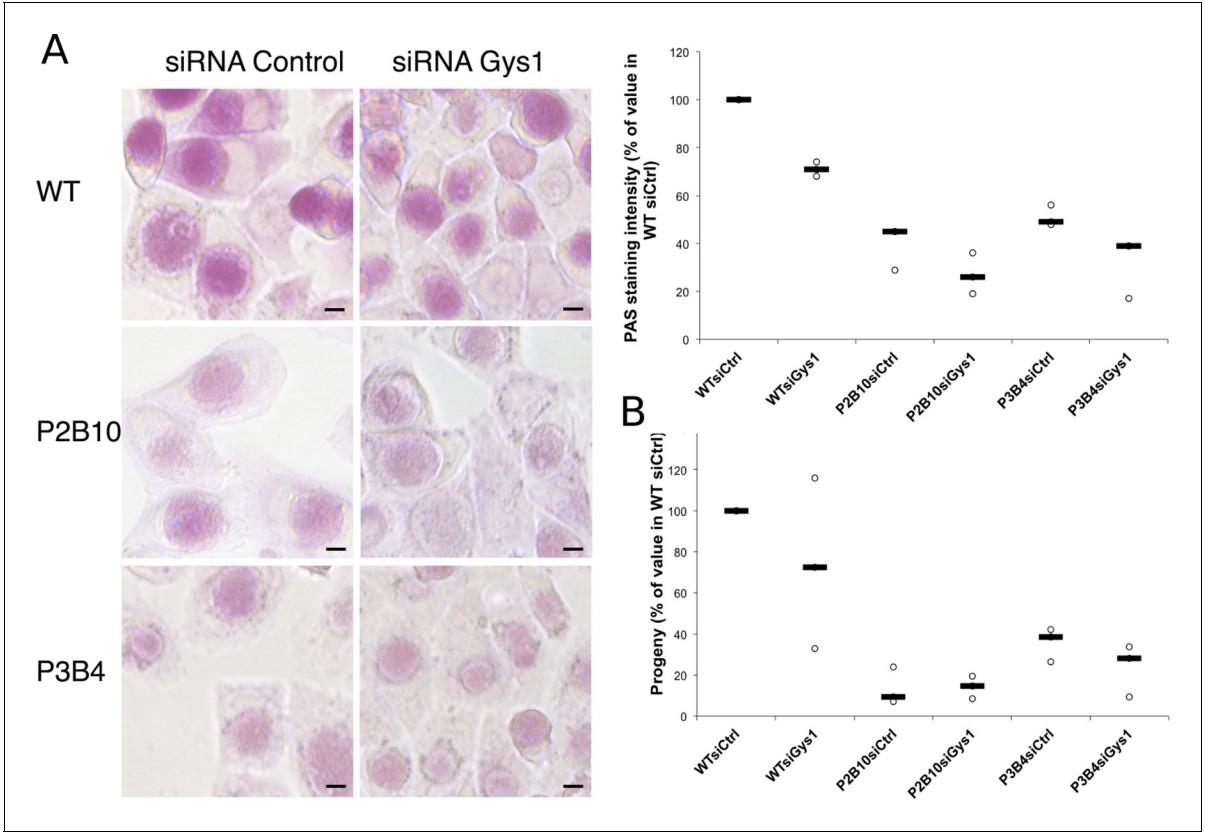

**Figure 7.** Mutations in GlgA result in defect in glycogen storage capacity and loss of infectivity. Cells were treated with either siRNA control or siRNA against Gys1 48 hr and 4 hr prior to infection with *glgA* mutants P2B10, P3B4 or with the parental wild-type strain. (**A**) PAS staining was performed 48 hpi. Pictures show representative fields for each strain. Mean intensity of glycogen staining in inclusions was quantified and expressed as a percentage of the mean value measured in cells treated with control siRNA and infected with the parental strain. The median of three independent experiments is shown. Mutations in GlgA resulted in reduced glycogen stores, which were decreased further upon depletion of Gys1. Scale bar: 10 µm. (**B**) Progeny was determined 48 hpi in a reinfection assay. Results are expressed as the percentage of progeny measured in cells treated with control siRNA and infected with the parental strain. The median of three independent experiments is shown.

The following figure supplement is available for figure 7:

**Figure supplement 1.** Alignment of GlgA sequences of different bacteria.

P3B4, which carry mutations in conserved residues of GlgA (H131Y for P2B10 and G386E for P3B4), suggesting that they may have a strong, if not complete, loss of function (*Figure 7—figure supplement 1*). Glycogen accumulation in P2B10 and P3B4 inclusions 48 hpi was reduced to 40 and 51% respectively of its level in the parental strain (*Figure 7A*). Impact on infectivity was assessed by collecting the bacteria 48 hpi, and re-infecting fresh HeLa cells. Progeny was reduced 7-fold and 3-fold in P2B10 and P3B4 respectively, demonstrating that a defect in GlgA results in an impaired infectivity in vitro (*Figure 7B*). However, it is impossible to discriminate if the deficiency is related to the defect in luminal storage of glycogen, or in glycogen synthesis in the bacteria, or both. We hypothesized that import of host Gys1 may partially compensate for the deficiency in secreted GlgA. In that case, depletion of Gys1 might affect luminal glycogen stores, and possibly decrease infectivity even further. In the parental strain, depletion of Gys1 reduced glycogen accumulation by 29%, similar to *C. trachomatis* reduction (*Figure 7A*). In the mutants, depletion of Gys1 also decreased glycogen accumulation, demonstrating that part of the residual glycogen observed in the mutants is due to bulk glycogen import. The fact that glycogen stores were not fully depleted suggests that the mutants we selected show reduced GlgA activity rather than complete loss of function. Interestingly, decreasing luminal glycogen further by silencing Gys1 expression content did not have a higher impact on the progeny of the mutants than it had on the parental strain, rather less so (*Figure 7B*). In particular, Gys1 depletion did not decrease progeny of the P2B10 mutant. This observation suggests that in this mutant, the defect in progeny is mostly due to a deficiency in bacterial glycogen synthesis rather than in luminal glycogen storage.

Altogether, these experiments confirm that glycogen accumulation in the inclusion lumen results from the activity of both host and bacterial glycogen synthases, the latter making the largest contribution. In addition, they demonstrate that impaired GlgA activity has a significant impact on bacterial infectivity.

## Discussion

This work shows that two independent processes contribute to glycogen accumulation in the *C. trachomatis* inclusion lumen: bulk uptake from the host cytoplasm and *de novo* synthesis (*Figure 8*). It is a unique example of a bacterium utilizing the compartmentalization of eukaryotic cells, to the

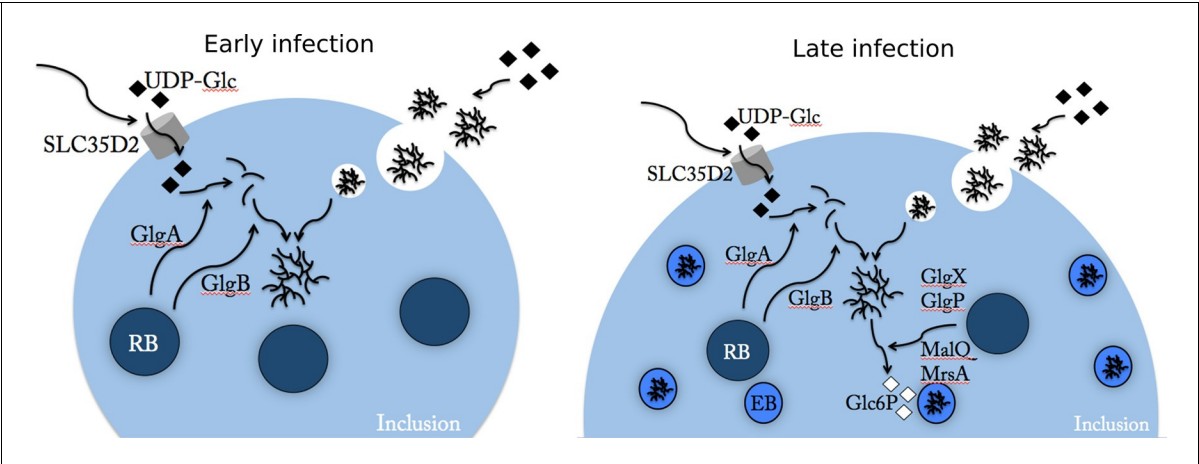

**Figure 8.** Glc flux in *C. trachomatis* infected cells. Early during the infectious cycle (left) the inclusion contains mostly RBs. This developmental form does not accumulate glycogen and uses ATP rather than Glc6P (*Omsland et al., 2012*). SLC35D2, and possibly other transporters, are recruited to the inclusion membrane and UDP-Glc is translocated into the inclusion lumen. The activity of chlamydial glycogen metabolism enzymes, secreted by RBs into the inclusion lumen, leads to the onset of luminal glycogen synthesis between 16 and 20 hpi. In addition, host glycogen is imported into the inclusion lumen through invagination of the inclusion membrane. In culture cells *de novo* synthesis predominates over bulk glycogen import. Later on (right), RBs start converting into EBs, which rely on Glc6P as energy source. EBs obtain Glc6P via the degradation of luminal glycogen into Glc1P, subsequently converted to Glc6P by the phosphoglucomutase (MrsA) and imported by UhpC. During RB to EB conversion T3S is turned off, allowing for intrabacterial activity of the glycogen metabolism enzymes, and glycogen accumulation in the bacteria.

extent that energy stores are radically shifted toward the bacterium and made inaccessible to the host.

Intraluminal glycogen accumulation had previously been thought to be due to bacterial lysis and to the release of chlamydial glycogen into the surrounding environment (*Chiappino et al., 1995*). In the present work we observed that glycogen appears first in the inclusion lumen and only later in EBs. Thus, bacterial lysis is not the source of luminal glycogen.

We identified two mechanisms by which glycogen accumulates in the inclusion lumen. First, the host enzyme Gys1, which is known to bind to glycogen (*Roach et al., 2012*), is translocated into the vacuole, arguing for uptake of host glycogen in bulk. The observation of glycogen-filled vesicles in the inclusion lumen also strongly argues for this scenario. Finally, depleting cytoplasmic glycogen by silencing Gys1 partially reduced luminal glycogen content in *C. trachomatis* and *C. muridarum*. Translocation of cytoplasmic glycogen confirms the ability for the inclusion to take up large particles from the host cytoplasm, already illustrated with the uptake of lipid droplets and peroxisomes (*Boncompain et al., 2014*; *Kumar et al., 2006*). This likely occurs through invagination of the inclusion membrane, but the underlying mechanisms remains to be investigated. While multi-layered glycogen-filled vesicles were observed, they still occurred in Atg5 deficient cells, implicating that they do not originate from an Atg5-dependent autophagic process.

The observation that depletion of host glycogen (through silencing of the host glycogen synthase Gys1) reduces luminal glycogen stores by no more than 20% demonstrates that bulk uptake is a minor pathway for glycogen accumulation in the inclusion lumen. It remains to be determined if this bulk import through a vesicular mechanism is glycogen-specific, or a general pathway used for the acquisition of a wide range of host components. We identified the second, and main, mechanism for luminal glycogen accumulation as *de novo* synthesis, through the action of chlamydial enzymes. Using a heterologous secretion assay, we have identified T3S signals in all but one (GlgC) glycogen metabolism enzymes of *C. trachomatis*. We had previously demonstrated the robustness of this assay, with an estimated 5% of false positives (*Subtil et al., 2005*). Secretion of GlgA in the inclusion lumen and in the host cytoplasm was reported recently (*Lu et al., 2013*). GlgA mutants of *C. muridarum* accumulated less glycogen in the inclusion lumen than their parental strain, demonstrating further that GlgA was indeed present and active in the inclusion lumen. Evidence for the secretion of the branching enzyme GlgB was brought by the observation that a GlgB mutant strain shows precipitation of unbranched glycogen in the inclusion (*Nguyen and Valdivia, 2012*). Finally in this paper, we show the secretion of GlgX using specific antibodies. Secretion of GlgX, and of other enzymes involved in glycogen metabolism (except GlgA), was not detected by an earlier study using mouse antisera (*Lu et al., 2013*). This study focused primarily on the detection of bacterial proteins in the host cytoplasm. Detection of the intraluminal pool, not overlapping with bacteria, might have been hindered by the signal coming from the pool of intra-bacterial enzymes. Finally, GlgX localizes not only to the inclusion lumen, but seems to be enriched at the inclusion membrane at 24 hpi. Interestingly, GlgP localization at the inclusion membrane was also reported (*Saka et al., 2011*), indicating that the two enzymes might work in concert, degrading host glycogen in the vicinity of the inclusion membrane. Thus, altogether these data indicate that *C. trachomatis* uses T3S to transform the lumen of the inclusion into a glycogen storage compartment.

So far, T3S is mostly described as a mechanism for protein translocation across a eukaryotic membrane, either the plasma membrane or the membrane of a parasitophorous vacuole (*Galan et al., 2014*). The ability for chlamydial glycogen enzymes to reach the inclusion lumen is intriguing. Loose membrane-like structures can frequently be seen in the inclusion lumen, and might trigger the secretion of some effectors into the inclusion lumen, especially from bacteria not in contact with the inclusion membrane. Also, it has been demonstrated, in *Yersinia pseudotuberculosis,* that T3S substrates can translocate first on the surface of the bacteria, without previous membrane contact (*Akopyan et al., 2011*). Alternatively, it may be that glycogen enzymes are first translocated into the cytoplasm, and transported back to the inclusion lumen directly or indirectly associated to glycogen, like Gys1. Clearly, more work is needed to understand how T3S is regulated in *Chlamydia*, and what determines substrate translocation across the inclusion membrane.

Our transcription analysis, in agreement with published data (*Belland et al., 2003*), indicates that *glgA* transcription most likely controls the onset of glycogen synthesis, as its initiation coincides with the onset of intraluminal glycogen accumulation (between 16 and 20 hpi). For all the other glycogen enzymes, transcription correlated with the known increase in bacterial metabolic activity between 8

and 24 hpi. The only other exception is *glgB*, which is one of the few early genes. It is not clear at this stage why the protein is made long before GlgA is present, when GlgA lies upstream of GlgB in the glycogen synthesis pathway. *glgB* is not in an operon, and its promoter region lies within a coding sequence, limiting the possibility of mutations in this region (*Albrecht et al., 2011*). Early expression of *glgB* might thus be a remnant of a different regulation of expression of the glycogen metabolism enzymes during the evolution of Chlamydiales.

To test the need for bacterial glycogen synthase activity in vitro we obtained two independent mutants in conserved residues of GlgA. Because of the mutagenesis strategy used, the mutants carried additional mutations (the library has an average of 16 mutations/genome). However, their similar phenotypes in respect to glycogen storage and infectivity strongly support the hypothesis that those are due to the GlgA mutation and not to additional genetic defects. These mutants demonstrated that defects in GlgA activity result in a loss of infectivity in vitro. We have not yet been able to obtain a stable strain knocked-out for *glgA* expression, but this might be due to technical difficulties rather than essentiality of the gene, and the question as to whether glycogen synthesis is required for chlamydial development remains open.

The very few *C. trachomatis* isolates that do not have the plasmid accumulate only minor amounts of glycogen compared to the wild-type strain, and GlgA expression is strongly reduced in these strains (*Carlson et al., 2008*). When we ectopically expressed Flag-GlgA in cells infected with either the wild-type or the plasmid-less strain, we observed increased intraluminal glycogen accumulation for both strains. These data show that different levels of GlgA expression between the two strains account for their difference in terms of glycogen accumulation. When Flag-GlgA import into the inclusion took place, likely together with bulk glycogen uptake, the low level of endogenous GlgA in the plasmid-less inclusions was complemented, allowing for high glycogen accumulation. Presumably, in the plasmid-less strain, bulk import of host glycogen occurs normally, but remains below detection. The fact that this strain has no infectivity defect in vitro (*Carlson et al., 2008*) indicates that high luminal glycogen accumulation is not needed for chlamydial growth in vitro. Still, the observation that Gys1 depletion decreased progeny by about 30% in the wild-type *C. muridarum* strain suggests that bacterial development is somewhat sensitive to the amount of glycogen in the inclusion lumen, maybe at early times of infection, before *glgA* is highly expressed and glycogen accumulation peaks. Alternatively, disruption of the glycogen storage capacity in the host might have indirect effects on nutrient balance, and account for the small decrease in infectivity.

In vitro polymerization assays using *E. coli* expressing *C. trachomatis* GlgA, as well as transfection experiments in a eukaryotic background where only UDP-Glc is available, proved that *C. trachomatis* GlgA can use UDP-Glc as substrate. This was unexpected, because eubacterial glycogen synthases normally use ADP-Glc. Consistent with this finding, we observed a strong decrease in intraluminal glycogen accumulation when we knocked down the human enzyme responsible for the generation of UDP-Glc (UGP2). Decrease of host glycogen levels (and thus the reduction in the bulk import pathway) could not account for this result, as the Gys1 knockdown was a lot more efficient in depleting glycogen in the host cytoplasm than the UGP2 knockdown. This experiment points to UDP-Glc as being the substrate imported into the inclusion lumen. If, as we had initially hypothesized, Glc6P or Glc1P were that substrate, a UGP2 knockdown should not produce any difference in intraluminal glycogen accumulation because both Glc6P and Glc1P would remain available, as they lie upstream of UDP-Glc generation. Energetically speaking, it is beneficial to import UDP-Glc rather than Glc6P or Glc1P, as it relieves the bacteria from the costly reaction of transferring a nucleotide to the sugar molecule. It also fits perfectly with the absence of a T3S signal in GlgC, suggesting that this enzyme, which makes ADP-Glc out of Glc1P, remains restricted to the bacteria and only serves bacterial glycogen production. Secretion of GlgC into the inclusion lumen would be superfluous, with GlgA being able to produce unbranched glycogen out of UDP-Glc.

The identification of UDP-Glc as the sugar imported into the inclusion lumen was further comforted by our observation that knocking down the UDP-Glc transporter SLC35D2 led to a significant reduction of intraluminal glycogen staining. In addition, we observed that SLC35D2-HA accumulates at the periphery of the inclusion. Our data thus strongly indicate that SLC35D2 is at least partially responsible for the import of UDP-Glc. The fact that the reduction was only partial, and slightly less pronounced than after UGP2 silencing, might be due to residual SLC35D2 expression. In addition, other Golgi- or ER-based transporters from the SLC35-family of nucleotide-sugar transporters might

transport UDP-Glc and be recruited to the inclusion membrane, accounting for the residual glycogen accumulation in the SLC35D2 knockdown.

We demonstrated that *de novo* glycogen synthesis takes place in the inclusion lumen, triggered by the presence of GlgA and GlgB. Glycogen storage in the inclusion lumen would only be of benefit for the bacteria if they were subsequently degraded into monomers amenable to bacterial uptake. The early observation that glycogen decreases at very late infection times, is consistent with a late consumption of the stores (*Gordon and Quan, 1965*). Similarly to the glycogen synthesizing enzymes, the degrading enzymes GlgP, GlgX and MalQ possess T3S signals, and we demonstrated GlgX secretion using specific antibodies. We therefore hypothesize that these enzymes are active in the inclusion lumen, and generate free Glc1P. We could clearly demonstrate that *Chlamydiae* are not able to take up Glc nor Glc1P, but exclusively Glc6P. This is in agreement with data obtained on the homologous protein in *C. pneumoniae*, which transports Glc6P and not Glc1P (*Schwoppe et al., 2002*). This apparent contradiction can be explained by the fact that the chlamydial phosphogluco-mutase MrsA (interconverting Glc1P and Glc6P) also possesses a T3S signal, and is thus most likely secreted. We have attempted to demonstrate MrsA secretion using newly developed tools to transform *C. trachomatis* (*Wang et al., 2011*). The C-terminal tagged MrsA was not expressed in transformed bacteria, neither with the endogenous promoter nor with a tetracyclin-inducible promoter (not shown). Thus, while we propose that intraluminal glycogen is degraded into Glc1P and is converted to Glc6P in the inclusion lumen, the involvement of bacterial MrsA in this conversion remains to be confirmed.

Altogether, our work reveals the origin of glycogen in the inclusion lumen and brings to light the complexity of the Glc flux in *C. trachomatis* infected cells (*Figure 8*, see legend for details). At a first glance, this complexity seems counter intuitive: why not import Glc6P directly from the host cell cytoplasm into the inclusion lumen and then directly into the bacteria? UDP-Glc import is initiated very early on, at a time when most bacteria are RBs, which, importantly, use ATP as an energy source, while EBs use Glc6P (*Omsland et al., 2012*). Uptake of Glc6P, before EBs appear, would dangerously increase the osmotic pressure in the inclusion. Stocking Glc in the shape of the osmotically inert glycogen brings an elegant solution. Why do *C. trachomatis*, amongst all the *Chlamydia* species, accumulate these high amounts of glycogen within their parasitophorous vacuole? Clearly, as discussed earlier, glycogen accumulation is not necessary for growth in vitro. Plasmid-less strains, that do not accumulate glycogen, were highly attenuated in a genital model of mouse infection and in ocular infection of non-human primates (*Carlson et al., 2008*; *Russell et al., 2011*). To date it is not known whether the lack of glycogen accumulation plays a role in this reduction in infectivity, or other traits associated with the absence of plasmid. Surprisingly, it was recently shown that the plasmid-less strain induced similar level of infection and pathologies as the wild-type strain in a model of genital infection in non-human primates (*Qu et al., 2015*). Thus, while, the ability for virtually all *C. trachomatis* clinical isolates to accumulate glycogen speaks for a selective advantage of doing so, we can only speculate on what this advantage might be. Many bacterial strains of the microflora of the female genital tract metabolize Glc as carbon source. Whether Glc can become limiting, especially during bacterial vaginosis, where the risk of becoming infected with *C. trachomatis* is elevated, is not known. It is an attractive hypothesis to explain why the ability for *C. trachomatis* to store Glc could come to an advantage in such specific conditions, which the macaque model would not reveal. One independent advantage of re-routing the host energy stores towards the inclusion lumen is that it might overall 'weaken' the host cell in its fight against the intruder. Indeed, many autonomous host cell defense mechanism rely on processes that require energy (including host protein synthesis or phosphorylation cascades) and that might be compromised in cells enduring sustained hijacking of its energy stores.

This work demonstrates that a microbe can deplete the host from its energy stores, making them available only to themselves. We propose that such sequestration of host molecular complexes, or even organelles, inside the parasitophorous vacuole, could be broadly used by other intracellular parasites, to ensure nutrient access and/or to disrupt defensive signalling pathways.

## Materials and methods

### Cells and bacteria

HeLa cells (ATCC), wild-type and *Atg5*-/- mouse embryonic fibroblasts (MEF) (generous gift from N. Mizushima, Tokyo Medical and Dental University) were cultured in Dulbecco's modified Eagle's medium with Glutamax (DMEM, Invitrogen, Carlsbad, CA), supplemented with 10% (v/v) fetal bovine serum (FBS). Cells were routinely checked for absence of mycoplasma contamination by PCR. For experiments with medium containing different Glc concentrations, DMEM without Glc (DMEM, Invitrogen) was used and complemented with 5 mM sodium pyruvate (Sigma-Aldrich), 10% fetal bovine serum (FBS) and the indicated Glc concentration (Merck). *C. trachomatis* LGV serovar L2 strain 434 (ATCC), the plasmid-less strain LGV L2 25667R (*Bowie, 1990*) or GFP-expressing L2 (L2$^{IncD}$GFP) (*Vromman et al., 2014*) were purified on density gradients as previously described (*Scidmore, 2005*). *Chlamydia muridarum* was a kind gift from Dr H Caldwell (Rocky Mountain Laboratories, NIAID, Hamilton, MT). McCoy mouse fibroblast cells (ATCC), and Vero monkey kidney cells (ATCC) were maintained and propagated as detailed in (*Rajaram et al., 2015*). The GlgB mutant was kindly provided by Dr R Valdivia (*Nguyen and Valdivia, 2012*). The *ipaB* and *mxiD* strains are derivates of M90T, the virulent wild-type strain of *Shigella flexneri*, in which the respective genes (*ipaB* and *mxiD*) have been inactivated (*Allaoui et al., 1993*). The *Escherichia coli* strain *DH5α* was used for cloning purposes and plasmid amplification. Both *S. flexneri* and *E. coli* strains were grown in Luria-Bertani medium with ampicillin at 0.1 mg/ml.

### Periodic acid-thiocarbohydrazide-silver proteinate reaction (PATAg)

HeLa cells were grown in wells, infected with *C. trachomatis* LGV serovar L2 strain 434 at an MOI of 0.1 and carefully trypsinized at the indicated time points. The cells were then washed with PBS once and fixed with 0.1 M cacodylate and 2.5% glutaraldehyde at room temperature for at least 30 min. PATAg staining was performed as described elsewhere (*Thiéry, 1967*). Briefly, thin sections were incubated in 1% periodic acid for 25 min and then washed several times in water, followed by an incubation step in 0.2% thiocarbohydrazide in 20% acetic acid for 45 min. Several washing steps in a graded acetic acid series to water were carried out and the thin sections were stained with 1% silver proteinate for 30 min. Samples were observed within a week after preparation. Images were obtained using a Tecnai T12 transmission electron microscope (FEI) at 100 kV and captured using an Eagle CCD camera (FEI).

### Electron microscopy

Transfected HeLa cells were fixed for 15 min at room temperature in 8% paraformaldehyde (PFA)/ 0.2% glutaraldehyde in PHEM buffer (60 mM Pipes, 25 mM Hepes, 10 mM EGTA and 2 mM MgCl$_2$, pH 7.2). Cells were carefully scraped off, centrifuged and fixation was allowed to continue for 45 min at room temperature with fresh fixative buffer before storage at 4°C. Then cells were subsequently washed three times with PHEM buffer and embedded in 12% gelatine in PHEM buffer at 37°C. Samples were centrifuged for 2 min at 20000 *g* and gelatine was solidified on ice for 20 min. Cubes (1 mm$^3$) were excised and impregnated overnight with 2.3 M sucrose at 4°C (*Tokuyasu, 1973*). Ultrathin sections (50–60 nm) were cut with a cryoultramicrotome (UC6/FC6; Leica Microsystems, Vienna, Austria). Those sections were collected with 0.575 M sucrose/0.05% (w/v) methylcellulose in PHEM buffer and dipped on to formvar/carbon-coated Ni grids. Immunolabelling was performed with mounted sections which were washed three times for 15 min with 25 mM NH$_4$Cl in PBS. The sections were blocked in 5% milk powder in PBS and subsequently incubated with a rat anti–HA antibody (clone 3F10 Roche Diagnostics) diluted at 1/50 in PBS-milk powder 5% for 30 min. The labelled sections were washed 6 times for 3 min in PBS-milk powder 0.5% and incubated for 30 min with a rabbit anti-rat IgG (Dako Denmark A/S, Glostrup, Denmark) diluted 1/200 in PBS-milk powder 0.5%. Sections were washed 6 times for 3 min with PBS milk-powder 0.5% and then incubated for 30 min with Protein-A gold-10 nm, diluted at 1/50 in PBS/0.5% BSA. The sections were then rinsed briefly 3 times and then 7 times for 3 min with PBS. The labelled sections were fixed for 5 min in PBS/1% glutaraldehyde and subsequently washed 8 times with distilled water. The sections were then washed twice with 0.4% aqueous uranyl acetate/1.8% (w/v) methylcellulose on ice and the incubation was

continued with fresh solutions for 5 min on ice. Sections were observed under a Tecnai T12 transmission electron microscope (FEI company) at 120 kV. Images were taken with an Eagle CCD camera.

## Immunofluorescence and PAS staining

HeLa cells grown on coverslips were infected with *C. trachomatis* LGV serovar L2 strain 434 with an MOI < 1 (unless specified differently) at 37°C and fixed in 4% PFA in PBS for 30 min at room temperature (except when staining with anti-CT813 was intended, which required fixation in 2% PFA). Cells were blocked and permeabilized in 0.05% saponin and 0.1% bovine serum albumin (BSA) in PBS for 10 min before being subjected to antibody staining. The antibody against inclusion protein CT813 was kindly provided by Dr. G. Zhong (San Antonio, Texas). Polyclonal anti-Cap1 antibodies were obtained by immunization of New Zealand white rabbits with purified recombinant Cap1 deleted of its last 167 amino acids and fused to a N-terminal His tag (Agro-Bio). The polyclonal anti-GlgX antibody was equally purchased from Agro-Bio, and was directed against two peptides (KHNEENGE YNRDGTSANC and HEDFDWEGDTPLHLPKEC). To investigate its specificity anti-GlgX was preabsorbed with either the two GlgX peptides or control peptides. For this, 1 µg/ml antibody was incubated with 20 µg/ml of peptides for 15 min at room temperature prior to immunostaining of cells. The monoclonal rat anti-HA antibody was purchased from Roche Diagnostics. Goat secondary antibodies were conjugated to Alexa488 (Molecular Probes), or to Cy3 or Cy5 (Amersham Biosciences). For periodic-acid-Schiff (PAS) stain cells grown on coverslips were fixed in 4% PFA/PBS for 30 min at room temperature and staining was performed as described (*Schaart et al., 2004*). Briefly, cells were incubated in 1% periodic acid (Sigma) for 5 min. Thereafter coverslips were put in tap water for 1 min, quickly rinsed in mQ-$H_2O$ and then applied to Schiff reagent (Sigma) for 15 min at room temperature. Afterwards the coverslips were rinsed again in mQ-$H_2O$, incubated in tap water for 10 min followed by an incubation step in PBS for 5 min. Periodic acid oxidizes the vicinal diols in sugars such as glycogen to aldehydes, which now in turn react with the Schiff reagent to give a purple/magenta colour. Images were acquired on an Axio observer Z1 microscope equipped with an ApoTome module (Zeiss, Germany) and a 63× Apochromat lens unless specified. Images were taken with a Coolsnap HQ camera (Photometrics, Tucson, AZ) using the software Axiovision.

## Glucose uptake assay

Gradient purified *Chlamydia* EBs were incubated in an axenic medium (5 mM $KH_2PO_4$, 10 mM $Na_2HPO_4$, 109.6 mM K-gluconate, 8 mM KCl, 1 mM $MgCl_2$) (*Omsland et al., 2012*) supplemented with 0.2 mM α-D-[$^{14}$C(U)]-Glc 1-Phosphate, α-D-[$^{14}$C(U)]-Glc 6-Phosphate or D-[$^{14}$C(U)]-Glc (Perkin Elmer) (0.1 µCi per sample). In some samples 10 mM of the indicated cold monosaccharide was added in a competition assay. After two hours of incubation at 37°C the bacteria were pelleted (15000 *g* for 5 min) and washed twice in 50 mM $K_2HPO_4$/$KH_2PO_4$, 100 mM KCl and 150 mM NaCl. Radioactivity associated to the bacterial pellet and to the supernatant was measured by a scintillation counter.

## Construction of recombinant plasmids

Genomic DNA from *C. trachomatis* LGV serovar L2 strain 434 was prepared from bacteria using the RapidPrep Micro Genomic DNA isolation kit (Amersham Pharmacia Biotech). The first 20 to 30 codons of different chlamydial genes including about 20 nucleotides upstream from the translation start sites were amplified by PCR using primers listed in *Supplementary file 1* and cloned into the pUC19cya vector as described (*Subtil et al., 2001*). attB-containing primers (*Supplementary file 1*, Gateway, Life technologies) were used to amplify and clone the *C. trachomatis* L2 *glgA* gene into a destination vector derived from the mammalian expression vector pCiNeo, providing an amino-terminal 3xflag tag, and into pDEST15 (Gateway).

## Transfection

siRNA transfections were performed with Lipofectamine RNAiMAX (Life technologies) according to the manufacturer's protocol. A mixture of 2 to 4 siRNA sequences (*Supplementary file 2*), with a final concentration of 10 nM each, was used and transfection was done twice, 48 hr and 4 hr prior to infection. siRNA efficiency was determined by immunoblot or RT-PCR (see respective sections for more detail).

Cells were transfected with plasmid DNA 24 hr after seeding using jetPRIME transfection kit (Polyplus transfection) according to the manufacturer's protocol. The constructs used were Flag-GlgA and SLC35D2-HA (pMKIT-neo-hRel8-cHA, kindly provided by Nobuhiro Ishida, Chiba Institute of Science, Japan) (Ishida et al., 2005).

## Reinfection assays

Cells were treated with the indicated siRNA 48 hr and 4 hr prior to infection, which was performed at an MOI < 0.3. Thirty or 48 hpi the cells were washed in PBS, detached and lysed with 1 mm glass beads. Fresh HeLa cells were inoculated with serial dilutions of the cell lysates and 24 hr later flow cytometry was used to determine the infection rate as described elsewhere (Vromman et al., 2014). Briefly, cells were washed with PBS and gently trypsinized. After a washing step in PBS cells were fixed in PFA 2% in PBS and bacteria were stained using FITC-coupled anti-Chlamydia antibody (Fitzgerald clone 502 #61R-1012). Infection rates were measured using a Gallios flow cytometer (Beckton Coulter).

## Quantification of glycogen with CellProfiler

The cell image analysis software CellProfiler was used to quantify glycogen content in inclusions stained with PAS. Around 50 inclusions were manually encircled, and their size and total staining intensity was determined. PAS staining not linked to the presence of glycogen was estimated by doing the same procedure on inclusions in cells grown in the absence of Glc. The averaged value obtained was considered background value and was substracted. Glycogen content of inclusions of cells treated with control siRNA was considered 100%.

## Quantitative reverse transcription PCR and reverse transcription PCR

Total RNA was isolated from $5 \times 10^5$ HeLa cells infected with *C. trachomatis* LGV serovar L2 after 1, 3, 8, 16, 24 hr or 40 hr of infection (MOI of 10 for 1, 3, 8 hr and MOI of 0.1 for 16, 24, 40 hr) with the RNeasy Mini Kit with DNase treatment (Qiagen) according to the manufacturer's protocol. RNA concentrations were determined by NanoDrop and the samples normalized to an equal RNA content. Reverse transcription (RT) was performed with SuperScript III Reverse Transcriptase (Life Technologies) and quantitative PCR (qPCR) undertaken with LightCycler 480 system using LightCycler 480 SYBR Green Master I (Roche). In parallel, genomic DNA (gDNA) of each time point was purified with the DNeasy Blood and Tissue Kit (Qiagen), and the amount of bacteria in the samples determined by qPCR using chlamydial primers. This was done to normalize the cDNA of the different samples. Primers are listed in *Supplementary file 3*, their specificity was ensured through the analysis of melting curves.

For the evaluation of siRNA SLC35D2 efficiency the steps until reverse transcription were the same, but the PCR was run with PrimeStar (Clontech). Equal volumes were loaded on agarose gels and bands were revealed using UV-light visualizing ethidium bromide.

## Western blot and antibodies

Cell pellets were lysed with a lysis buffer (8 M urea, 30 mM Tris, 150 mM NaCl, 1% v/v sodium dodecyl-sulfate) and proteins were subjected to sodium dodecyl-sulfate poly-acrylamide gel electrophoresis (SDS-PAGE) and transferred to a polyvinylidene difluoride (PVDF) membrane, which was blocked with 1× PBS containing 5% milk and 0.01% Tween-20. The membranes were then immunoblotted with primary antibodies diluted in 1 x PBS and 0.01% Tween-20. Primary antibodies used in the secretion assay were mouse anti-cya, rabbit anti-CRP and rabbit anti-IpaD and were generously given by Drs N Guiso, A Ullmann and C Parsot, respectively (Institut Pasteur, Paris). Other antibodies used were rabbit anti-Gys1 (Millipore #04–357), rabbit anti-UGP2 (GeneTex #GTX107967), mouse anti-Flag M2 (Sigma-Aldrich), goat anti-mouse IgG coupled to horseradish peroxidase (HRP) and goat anti-rabbit IgG-HRP (GE Healthcare). Blots were developed using the Western Lightning Chemiluminescence Reagent (GE Healthcare).

## Zymogram

Glycogen synthase genes (*glgA*) were amplified from genomic DNA of *Escherichia coli* K12 and *Chlamydia trachomatis* D/UW-3/CX and cloned into the expression vector pGEX (GE Healthcare) and pDEST15, respectively.

Precultures of wild-type and transformed *E. coli* were grown overnight in LB medium at 37°C, then transferred to fresh LB medium and grown until the optical density (OD) 600 nm reached 0.6. Cultures were subsequently induced overnight with 0.5 mM of IPTG.

Cells were harvested by centrifugation and disrupted by a French press at 1250 psi. The lysate was centrifuged at 16,000 $g$ for 15 min at 4°C. Soluble proteins (such as GlgA) were found in the supernatant. Glycogen synthase activity was detected by zymogram analysis. The proteins in the supernatant were separated by non-denaturing polyacrylamide gel electrophoresis (PAGE) containing 0.6% rabbit glycogen (Sigma-Aldrich). After electrophoresis, gels were incubated overnight at room temperature in glycogen synthase buffer (66 mM glycyl-glycine, 66 mM $(NH_4)_2SO_4$, 8 mM $MgCl_2$, 6 mM 2-mercaptoethanol, and 1.2 mM ADP-Glc or UDP-Glc). Glycogen synthase activity was then visualized as dark activity band after iodine staining.

## Heterologous secretion assay in *Shigella flexneri*

Analysis of secreted proteins was performed as described elsewhere (*Ball et al., 2013*; *Subtil et al., 2001*). Briefly, 1 ml of a 30°C overnight culture of *Shigella flexneri ipaB* or *mxiD* transformed with different Cya chimeras was inoculated in 30 ml of LB broth with 0.1 mg/ml ampicillin and incubated at 37°C for 4 hr. Bacteria were then harvested by centrifugation and the supernatant was filtered through a Millipore filter (0.2 µm). To precipitate the proteins 1/10 (v/v) of trichloroacetic acid was added to the supernatant and the precipitate as well as the bacterial pellet resuspended in sample buffer for analysis by SDS-PAGE and immunoblot.

## EMS mutagenesis and library construction

*C. muridarum* was mutagenized by treating infected Vero cells with 15 mg/ml ethyl methanesulfonate (EMS) for 1 hr, starting 16 hpi, resulting in an average of 16 mutations/genome with a range of 5–26 mutations/genome (*Rajaram et al., 2015*). Mutagenized *Chlamydia* EBs were harvested at 26 hpi in sucrose phosphate glutamic acid buffer (SPG) using 3 mm glass beads, and aliquots were stored at -80°C. Mutant libraries were constructed as described previously (*Rajaram et al., 2015*). Briefly, mutagenized *C. muridarum* isolates were plaque-cloned in McCoy cells and plaques were picked 10 days post infection. Isolated plaques were expanded in McCoy cells by infecting monolayers in 96-well plates with 30 µl of plaque lysates by centrifugation (1600 $g$ for 1 hr at room temperature) and rocking (30 min at 37°C). The library plates were stored at -80°C in SPG.

## Isolation of glycogen deficient mutants

Five µl of *C. muridarum* mutant isolates from the mutant library were used to infect fresh McCoy cell monolayers by centrifugation and rocking. At 28 hpi monolayers were fixed with 100% methanol for 10 min. Glycogen was assayed by staining with a 5% iodine solution (5% w/v iodine (Sigma), 5% w/v potassium iodide (Sigma), in a 50/50 solution of water and 100% ethanol) for 10 min, followed by staining with a 2.5% iodine solution (2.5% w/v iodine, 2.5% w/v potassium iodide, in a 50/50 solution of water and 100% ethanol) for 10 min. The iodine solution was removed and 100 µl PBS was added to each well. Plates were imaged using an EVOS® FL Auto cell imaging microscope. Glycogen deficient mutants were examined for mutations in known glycogen biosynthesis genes by Sanger sequencing or by whole genome sequencing as previously described (*Rajaram et al., 2015*). P2B10 carried a C393T mutation in *glgA*, and P3B4 carried a G1157A mutation in the same gene.

## Acknowledgements

We thank Dr. A Dautry for critical reading of the manuscript. We are thankful to people who contributed tools and reagents: I Clarke (University of Southampton, UK), N Mizushima (Tokyo Medical and Dental University, Japan), Nobuhiro Ishida (Chiba Institute of Science, Japan), and R Valdivia (Duke University Medical Center, USA). This work was supported by an ERC Starting Grant (NUChLEAR

N°282046), the ANR (Ménage à trois, ANR-12-BSV2-0009 and Expendo ANR-14-CE-0024), the Institut Pasteur and the Centre National de la Recherche Scientifique.

## Additional information

### Funding

| Funder | Grant reference number | Author |
|---|---|---|
| Institut Pasteur | | Lena Gehre<br>Olivier Gorgette<br>Stéphanie Perrinet<br>Marie-Christine Prevost<br>Agathe Subtil |
| Centre National de la Recherche Scientifique | | Lena Gehre<br>Mathieu Ducatez<br>Steven G Ball<br>Agathe Subtil<br>Stéphanie Perrinet |
| Agence Nationale de la Recherche | Menage a trois ANR-12-BSV2-0009 | Lena Gehre<br>Mathieu Ducatez<br>Steven G Ball<br>Agathe Subtil |
| European Research Council | NUChLEAR_282046 | Lena Gehre<br>Agathe Subtil |
| Agence Nationale de la Recherche | Expendo ANR-14-CE-0024 | Mathieu Ducatez<br>Steven G Ball<br>Agathe Subtil |

The funders had no role in study design, data collection and interpretation, or the decision to submit the work for publication.

### Author contributions

LG, Performed the experiments, Interpreted the data, Wrote the article; OG, Performed PATAg, Acquired pictures, Quantified the data; SP, Provided technical assistance for several of the siRNA experiments, PAS stainings, reinfection assays, Performed the experiments with *C. muridarum*; M-CP, Performed PATAg and immunogold labelling on samples, Acquired pictures, Quantified the data; MD, Performed in vitro assays on GlgA activity; AMG, Obtained the GlgA mutants in *C. muridarum*, Contributed to writing the manuscript; DEN, Obtained the GlgA mutants in *C. muridarum*, Contributed to revising the manuscript; SGB, Contributed to the conception and design of the experiments, to the analysis and interpretation of data and to writing the article; AS, Conceived and designed the project, Contributed to data acquisition and analysis, Wrote the article

## Additional files

### Supplementary files

• Supplementary file 1. Primers used for cloning purposes.

• Supplementary file 2. List of siRNAs.

• Supplementary file 3. List of primers used in qRT-PCR and RT-PCR.

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
