## [Decision Letter]

Thank you for submitting your work entitled "Sequestration of host metabolism by an intracellular pathogen" for consideration by *eLife*. Your article has been favorably evaluated by three reviewers, including Rey Carabero and Dominique Soldati-Favre, who is a member of our Board of Reviewing Editors. The evaluation was overseen by Richard Losick as the Senior Editor.

The reviewers have discussed the reviews with one another and the Reviewing Editor has drafted this decision to help you prepare a revised submission.

Summary:

In this manuscript, the authors investigate how *Chlamydia trachomatis* ensures the establishment of a glycogen-rich vacuole by sequestering host cell metabolites and providing the necessary enzymes. They determined two parallel sources, which are differentiated by the timing of their requirement relative to the infectious cycle. At early stages of development, bulk import predominates, whilst later in infection with increasing differentiation to elementary bodies, de novo glycogen synthesis mediated by bacterial enzymes takes over. By defining the source the sugar species utilized – the translocation of the relevant enzymes by type III secretion – the report makes a significant contribution to chlamydial biology, and possibly pathogenesis in regards to nutritional immunity.

The present study is important as it highlights several unique and unappreciated phenomena:

1) Transport of a host biosynthetic enzyme into the inclusion.

2) de novo synthesis of glycogen in the inclusion through the secretion of bacterial biosynthetic enzymes.

3) Adaptation of a bacterial enzyme (GlgA) to use host-derived substrates (UDP glucose) for the synthesis of glycogen.

4) Recruitment of a host UDP glucose transporter to the inclusion and evidence that this transport activity is required for glycogen synthesis in the vacuole.

Essential revisions:

If glycogen is an important source of glucose6P, which the authors assume is being generated by the bacterial MrsA from glucose1P, then one would expect a phenotype for the resulting EBs. As it stands the study does not make full use of available technologies to move beyond correlation to causation, which are now feasible in *Chlamydia*. Mutants defective in *glg* genes are available – or can be generated readily by group II mediated retro-insertion (e.g. Targetron) to formally test the role of glycogen in *Chlamydia* growth and infectivity and glucose6p acquisition by EBs.

---

## [Author Response]

Essential revisions: If glycogen is an important source of glucose6P, which the authors assume is being generated by the bacterial MrsA from glucose1P, then one would expect a phenotype for the resulting EBs. As it stands the study does not make full use of available technologies to move beyond correlation to causation, which are now feasible in Chlamydia. Mutants defective in glg genes are available – or can be generated readily by group II mediated retro-insertion (e.g. Targetron) to formally test the role of glycogen in Chlamydia growth and infectivity and glucose6p acquisition by EBs.

The suggestion to use available technologies to generate mutants on some of the key enzymes of glycogen metabolism, and use those to “formally test the role of glycogen in Chlamydia growth and infectivity and glucose6p acquisition by EBs”, is at first thought quite attractive. However, obtaining such mutants is still notoriously laborious and could take several months, well in excess of the couple of months required for the revision procedure.

Nevertheless, we agree with the reviewers that by using mutants we could ask whether a defect in bacterial glycogen synthase activity has any effect on infectivity. To respond as fast as possible to the reviewers’ concerns, we established a collaboration with David Nelson (Indiana, USA) who had obtained GlgA mutants in *C. muridarum* (unpublished). The two GlgA mutants we tested accumulated less glycogen in the inclusion lumen than their parental strain, bringing further evidence that GlgA was indeed active in the inclusion lumen. The mutants also showed a defect in infectivity in vitro(3 and 7-fold respectively), bringing the finale evidence that the capacity for *Chlamydia* to metabolise glucose into glycogen is important for the completion of their developmental cycle.

In addition, we addressed the effect of Gys1 depletion on *C. muridarum*, and on the GlgA mutants. For a comparison, these experiments were done in parallel with *C. trachomatis* infections, and thereby we obtained sufficient data for *C. trachomatis* to detect a small but significant 20% decrease in luminal glycogen stores when Gys 1 was depleted (and 30% decrease in *C. muridarum*). This result brings further support to the “bulk” import pathway and the manuscript has been slightly modified accordingly (in the initial submission our dataset was too small to make the 20% decrease significant).

All the results obtained with the GlgA mutants were included at the end of the Results section, and are shown in a novel Figure 7. The question as to whether GlgA is absolutely required for bacterial development, and the effect of the *glgA* mutations in an in vivo model of infection, will be addressed in a separate report (Giebel et al. in preparation).

We acknowledge the interest of investigating mutants affecting those enzymes that are part glycogen of metabolism per se (*glgA, glgB, glgP, glgX, malQ, glgP*) and have responded to the suggestion by investigating defects in the genes encoding the major player of our story: GlgA. However, it is not clear to us what could be concluded from looking at a mutant in the phosphoglucomutase activity (MrsA). Indeed, phosphoglucomutase activity is implicated in all biochemical steps that require Glc1P, since *Chlamydiaceae* lack a hexokinase and only import Glc6P. This will include synthesis of intrabacterial glycogen (G1P is the substrate of GlgC, MrsA activity in the bacteria is needed to make bacterial glycogen out of imported G6P, independently of its potential role in the inclusion lumen). It might also include synthesis of chlamydial lipooligosaccharide. Indeed, phosphoglucomutase activity is needed for LPS synthesis in a bacterial symbiont of plants (Lepeck Molecular Plant Microbe Interactions 2002 15 368). If this holds true in *Chlamydia*, we can expect a very strong phenotype on progeny since LPS is required for RB to EB conversion (Nguyen PNAS 2011). Thus, pleiotropic effects of *mrsA* disruption would likely preclude testing our hypothesis that MrsA is involved in G1P to G6P conversion in the lumen of the inclusion. Although we agree that sophisticated schemes of complementation of a putative MrsA mutant could be attempted to test our hypothesis, we believe that obtaining mutants, and testing complementation experiments, need to be left to a follow-up study.